# Dynamics of the charge transfer to solvent process in aqueous iodide

Jinggang Lan [1,2,3] ✉, Majed Chergui [4,5] & Alfredo Pasquarello [1]

Charge-transfer-to-solvent states in aqueous halides are ideal systems for studying the electron-transfer dynamics to the solvent involving a complex interplay between electronic excitation and solvent polarization. Despite extensive experimental investigations, a full picture of the charge-transfer-to-solvent dynamics has remained elusive. Here, we visualise the intricate interplay between the dynamics of the electron and the solvent polarization occurring in this process. Through the combined use of ab initio molecular dynamics and machine learning methods, we investigate the structure, dynamics and free energy as the excited electron evolves through the charge-transfer-to-solvent process, which we characterize as a sequence of states denoted charge-transfer-to-solvent, contact-pair, solvent-separated, and hydrated electron states, depending on the distance between the iodine and the excited electron. Our assignment of the charge-transfer-to-solvent states is supported by the good agreement between calculated and measured vertical binding energies. Our results reveal the charge transfer process in terms of the underlying atomic processes and mechanisms.

Several inorganic anions, when dissolved in polar solutions, exhibit broad ultraviolet absorption bands, such as those observed in aqueous iodide at 225 and 193 nm, attributed to what are known as charge-transfer-to-solvent (CTTS) states. These CTTS states represent unique interactions between the solute and the solvent, forming quasi-bound states not observed in isolated ions. The valence electrons of ground-state anions are bound by the nucleus, but the excited states are bound by solvent polarization[1]. Although CTTS states also occur in molecular ions[2,3], atomic anions and, in particular, aqueous halides have received much attention from the ultrafast spectroscopy community as they are ideal systems for studying the electron-transfer dynamics to the solvent, since the solute lacks internal degrees of freedom. Indeed, the CTTS states of aqueous halides decay at ultrafast timescales, ejecting an electron into the solvent and leaving a neutral halogen atom behind[4–11]. In this process, electron ejection is entirely governed by the initial structure and motions of the molecules in the solvation shell[12,13].

Understanding the CTTS dynamics and the way the nascent solvated electron arises is critical for a description of pure electronic solvation, especially in a solvent as important as water, which is the medium of numerous biochemical reactions involving charge and energy transfer reactions[14–17]. It is, therefore, no surprise that the CTTS dynamics have been investigated by a wide variety of experimental methods, such as ultrafast fluorescence spectroscopy[18], transient visible and infrared absorption spectroscopy (TAS)[1,4,7,19], time-resolved photoelectron spectroscopy (TRPES)[8,20–22], etc. More recently, ultrafast X-ray absorption spectroscopy has been used to identify the fate of the nascent neutral halogen, although these studies are carried out using multiphoton excitation of the iodide, which leads to its direct ionization, rather than to the excitation of the CTTS state[10,11,23]. Among experimental methods using direct excitation of the CTTS state, TAS has been the most used approach, but it is largely dominated by the signal of the electron, which possesses a giant oscillator strength and is characterised by a very broad band extending from the ultraviolet to

[1]Chaire de Simulation à l'Echelle Atomique (CSEA), Ecole Polytechnique Fédérale de Lausanne (EPFL), CH-1015 Lausanne, Switzerland. [2]Department of Chemistry, New York University, New York, NY 10003, USA. [3]Simons Center for Computational Physical Chemistry at New York University, New York, NY 10003, USA. [4]Lausanne Centre for Ultrafast Science (LACUS), ISIC, Ecole Polytechnique Fédérale de Lausanne (EPFL), CH-1015 Lausanne, Switzerland. [5]Elettra - Sincrotrone Trieste, Area Science Park I - 34149, Trieste, Italy. ✉e-mail: jinggang.lan@nyu.edu

the mid-infrared. A spectroscopic fingerprint of the CTTS state has never been reported in TAS. Ultrafast fluorescence monitors the CTTS dynamics, because the CTTS emits light as long as there is still an overlap between the excited and ground-state wave functions. TRPES can also monitor the CTTS dynamics by virtue of its ability to measure the binding energy of the excited state and the subsequent steps.

Alternatively, theory has been used to simulate the CTTS process. This requires consideration of the solvent dynamics and a quantum-mechanical description of the solute and of the excited electron. The pioneering studies of Sheu and Rossky[24–26] and of Staib and Borgis[27–29] used mixed quantum-classical simulations based on one-electron pseudopotential models to reveal charge separation from the excited halide anion. While these simulations can provide valuable insight into the reaction mechanism, they do not fully account for the quantum nature of the solute and the solvent. This limitation can be addressed by using many-electron quantum chemistry simulations, which handle the solute-solvent interactions more accurately. However, such simulations have generally been restrained to the study of $I^-(H_2O)_n$ clusters[10,30–33] where only a few water molecules are taken into account. It is important to note that water clusters are not necessarily representative of the realistic water solvent and may not capture the full complexity of the solvent environment. Only recently, Carter-Fenk et al. obtained an ab initio molecular dynamics trajectory of a single CTTS event treating the spin singlet excitation in the restricted open-shell Kohn-Sham approximation and providing in this way the first glance at this process in a realistic solvent environment[34]. Despite the extensive experimental and theoretical work conducted on the CTTS process of iodide in water[1,4,7,8,14–17,19,21,22,24–36], critical questions remain unanswered concerning the structure of the CTTS state, its early time evolution prior to electron release, and the localization of the latter with respect to the solute still remain to be clarified. Specifically, the ultrafast UV fluorescence data[18] need to be reconciled with the TRPES ones[8,22]. In addition, there is an on-going debate about the occurrence and nature of contact pairs between iodine and the electron[5,8].

Here, we acquire a complete view on the interplay between electron and solvent evolution occurring during the CTTS process in aqueous iodide. The combined use of ab initio molecular dynamics and machine learning methods gives access to a full statistical description enabling multiple simulations and long time scales. We analyze the evolution of the excited electron during the CTTS process through its structure, dynamics, and free-energy landscape. This process is categorized into distinct states: CTTS, contact-pair, solvent-separated, and hydrated electron states, as determined by the iodine-electron distance. This assignment is shown to be in excellent agreement with measured vertical binding energies. In this work, we elucidate the charge transfer process in terms of its underlying atomic processes and mechanisms.

## Results

To accurately model the dynamics of the CTTS process, we combine advanced electronic structure theory, path-integral molecular dynamics, and machine learning methods. The underlying electronic structure is obtained with a van der Waals corrected hybrid functional[37,38], which correctly captures the electronic, structural, and dynamical properties of liquid water and of the hydrated electron[39–42]. We perform three sets of simulations in order to capture the dynamics of the CTTS process: (i) aqueous iodide in the singlet ground state ($I^-_{aq}$), (ii) aqueous iodine with an excited electron in the triplet state ($I_{aq} + e^-_{aq}$), and (iii) aqueous iodine radical ($I_{aq}$). All simulations have been performed with a single iodine atom and 127 water molecules at room temperature. For (i) and (iii), we perform long-timescale (50 ps) simulations of aqueous iodide and aqueous iodine with a machine learning potential. In (ii), the excited state is simulated using the unrestricted Kohn-Sham method with a triplet ($T_1$) state, which is in good agreement with the singlet excited state ($S_1$) as obtained from

time-dependent density functional theory calculations[43], cf. benchmark in the Supplementary Information (SI). The hybrid-functional-based molecular dynamics trajectory is accelerated through a multiple-time step scheme[44]. We carry out five simulations starting from equilibrated geometries of aqueous iodide. Among these simulations, four runs have a duration of 5 ps, and one lasts 25 ps. To obtain more statistics for the CTTS state, we also perform 50 independent molecular dynamics runs lasting 100 fs each. All simulations described above have been repeated with quantum nuclei combining the multiple-time step[44] and ring polymer contraction schemes[45,46]. The free-energy profile for the excited electron as a function of distance from the iodine is obtained through blue-moon sampling[47]. For the excited electron, we obtain the density of the Kohn-Sham orbital as $|\psi_{ks}|^2$ and calculate the corresponding gyration center and radius.

### Structural properties

When iodide is dissolved in water, the negative charge of the ion perturbs the neighboring hydrogen network of water, leading to a rearrangement of the water structure at the solute-solvent interface. The hydration shell of $I^-_{aq}$ is characterized by the radial distribution functions $g_{IO}$ and $g_{IH}$, cf. Fig. 1a, b. Nuclear quantum effects systematically broaden the features in the radial distributions without causing major structural modifications. This might stem from a near-perfect compensation of opposing quantum effects, as often remarked in previous literature[42,48]. The $g_{IO}$ and $g_{IH}$ obtained from our simulations reveal broad first peaks with maxima at 3.50 and 2.60 Å, in agreement with respective I-O and I-H distances of 3.50 and 2.65 Å derived from X-ray absorption fine structure measurements and previous simulations[49–52]. The resulting structure of $I^-_{aq}$ involves approximately seven water molecules, as inferred from the integration of $g_{IO}$ up to 4 Å for both classical and quantum nuclei. As shown in Fig. 1c, the orientation of the water molecules around the iodide ion is broadly distributed, but in the most common configuration the OH group points towards the iodide ion primarily driven by the electrostatic interaction between the charge and the dipole[50–52].

To investigate the CTTS state, we consider 50 independent molecular dynamics simulations of 100 fs and calculate the corresponding radial distribution functions, cf. Fig. 1d, e. Similar to the case of ground-state iodide, nuclear quantum effects only lead to a broadening of the structural features in the radial distribution functions. We analyze these trajectories in two time periods, namely 0–50 fs and 50–100 fs, in order to emphasize the structural evolution, cf. Fig. 1d, e. Iodide is isoelectronic with Xe, and therefore the CTTS state is similar to a diffuse Rydberg state, whose orbital leads to the repulsion of the surrounding molecules due to Pauli exclusion[53].

The transition from iodide to iodine initiates the readjustment of the water hydration shell around the I atom. In their experimental study, Suzuki et al. monitored the transition from the CTTS state to the contact-pair state[21]. In the latter state, both the iodine and the excited electron reside within the same solvation shell prior to its rearrangement. A significant variation in the time constant of this transition was observed upon the substitution of $H_2O$ with $D_2O$, corresponding to a change by a factor of 1.5. This isotope effect was ascribed to a solvent response dominated by atomic hydrogen motions. Specifically, the excited electron was suggested to attract hydrogen atoms, thereby causing the reorientation of the water molecules during this early phase of the CTTS reaction. Indeed, the $g_{IO}$ associated with the CTTS state (Fig. 1d) is comparable to that of aqueous iodide (Fig. 1a), except for a slightly lower and broader first peak. Due to their larger mass, the oxygen atoms are not able to respond on these short time scales, despite the dramatic change of the electronic configuration from iodide to iodine. In contrast, the first peak of $g_{IH}$ dramatically decreases, indicating that the hydrogen atoms rapidly reposition (Fig. 1e). The fast reorientation of the water molecules is evident in Fig. 1f, which indicates that the OH group reorients away from the

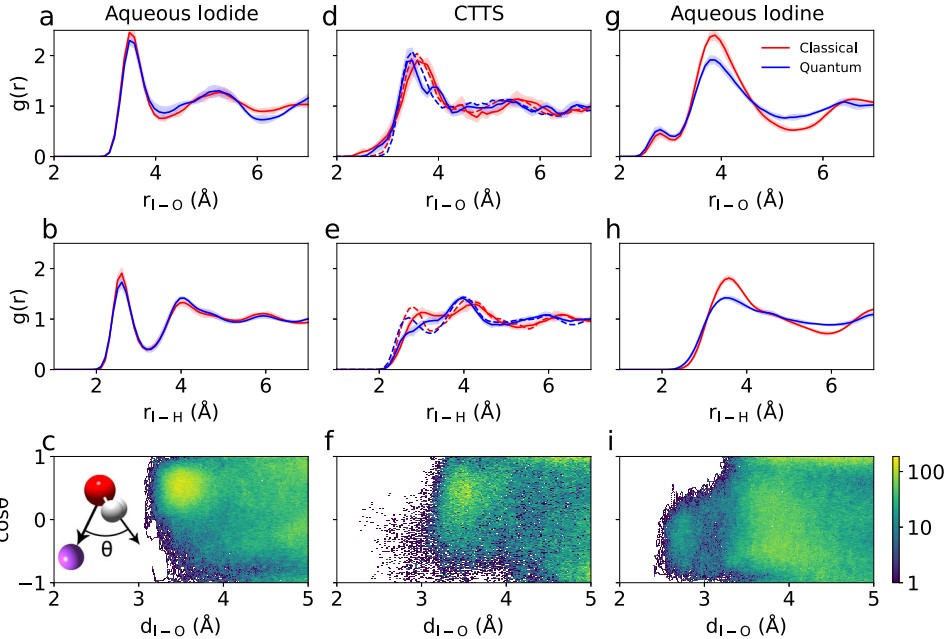

**Fig. 1 | Structural properties of CTTS dynamics.** The radial distribution function of (**a**, **d**, **g**) oxygen and (**b**, **e**, **h**) hydrogen with respect to the iodide/iodine solute for various states, as obtained with ab initio molecular dynamics simulation with classical (red) and quantum (blue) nuclei. **a**, **b** Aqueous iodide obtained from a 50-ps long trajectory; **d**, **e** CTTS state as obtained from 50 different 100 fs long trajectories starting from random initial structures of aqueous iodide, averages over the period from 0 to 50 fs (dashed-line) and over the period from 50 to 100 fs (solid-line) are given separately; **g**, **h** aqueous iodine as obtained from a 50-ps trajectory. Error bars correspond to standard error of mean calculated using block averaging. **c**, **f**, **i** $\cos\theta$ distribution from simulations with quantum nuclei, where $\theta$ is the angle between the water dipole and the oxygen-iodine vector as shown in the inset in **c**, for **c** aqueous iodide, **f** CTTS (0–100 fs), and **i** aqueous iodine. Positive values indicate that the water dipole points towards the iodine atom, while negative values indicate outward orientation. As the color changes from blue to yellow, the weight of the $\cos\theta$ distribution increases.

iodine atom towards the solvent. This effect becomes even more evident with time.

At longer times after excitation, the solvated electron has departed from the iodine atom. The latter can then be considered as an isolated iodine atom solvated in water. To model this limiting configuration, we investigate its hydration structure considering a 50-ps molecular dynamics trajectory (Fig. 1g–i). For aqueous iodine, both radial distribution functions $g_{IO}$ and $g_{IH}$ present distinct features that distinguish them from those of iodide and of the CTTS state. Compared to iodide, the first peak in $g_{IO}$ splits into two peaks, with the first peak located at 2.72 Å and the second peak at 3.85 Å, whereas $g_{IH}$ features a single peak at ∼3.6 Å. The first peak in $g_{IO}$ involves a single water molecule that occurs at a noticeably smaller distance than the main peak for iodide, indicating the formation of a well-defined iodine-water complex. The water molecule forming the complex is bound to the iodine through its oxygen atom, attracted by the hole in the outer electronic shell of $I_{aq}$. This orientation of the inner water molecule is not perturbed by the occurrence of an excited electron at nearby distances (4–6 Å from $I_{aq}$), as seen in simulation (ii) (see Supplementary Fig. 8). We remark that the nuclear quantum effects affect the second solvation shell at ∼4 Å, because the structural modifications within the $I^--H_2O$ complex substantially alter the hydrogen bond network of the surrounding water molecules. The formation of an $I^--H_2O$ complex was also reported in the quantum-mechanical/molecular mechanics (QM/MM) molecular dynamics (MD) simulations by Pham et al. and Penfold et al.[10,51] and was supported by the femtosecond X-ray absorption spectra presented therein. The other water molecules shift to larger distances, leading to a coordination number as large as 20 when integrating $g_{IO}$ up to its local minimum at 5.25 Å, significantly higher than the coordination number of 7 found in the case of iodide. This is also consistent with the results of ref. 10, where the coordination number of oxygen atoms around iodine was found between 20 and 25. These observations are attributed to the transition from a hydrophilic solvation around iodide to a hydrophobic solvation around iodine upon electron abstraction from iodide by multiphoton excitation[10].

## Dynamical properties

Next, we focus on the dynamics leading to the formation of the aqueous electron ($e^-_{aq}$) upon excitation. Building on the state assignment method proposed by Suzuki et al.[21,22], we adopt the distance between the iodine atom and the excited electron as a criterion to distinguish the states through which the system sequentially evolves: CTTS (Fig. 2a,b), contact-pair (Fig. 2c), solvent-shared (Fig. 2d), solvent-separated (Fig. 2e), and hydrated electron far from the aqueous iodine (Fig. 2f). With the exception of the solvent-shared state, this classification corresponds to that introduced by Suzuki et al.[21,22]. To rationalize our assignment, we analyze the dynamics of the electron density immediately upon excitation. The electron state is then characterized by rather large gyration radii $r_{gyr}$ of ∼4.5 Å (Fig. 2i) and small iodine-electron distances $d_{I-e}$ peaking at 1.2 Å (Fig. 2h). For the 50 independent CTTS trajectories, we obtain the time evolution of both $d_{I-e}$ and $r_{gyr}$ during the first 100 fs upon electron excitation (See Supplementary Fig. 9). We observe that the majority of electrons localizes in a time frame of 50–100 fs showing an average gyration radius of 3.0 Å. The electron and the iodine are then in contact and share the same solvation shell, which we characterize as the contact-pair state (Fig. 2c). This state corresponds to $d_{I-e}$ distances between 3.3 to 4.3 Å (orange region in Fig. 2h). Concurrently, we assign the CTTS state to $d_{I-e}$ lower than 3.3 Å (red region in Fig. 2h). As the dynamics develop further, we encounter the solvent-shared state, in which a single water molecule is shared by the aqueous iodine and the hydrated electron (Fig. 2d). This state occurs in a narrow range of distances at 4.3 ± 0.1 Å. Next, two separate solvation shells form giving rise to the solvent-separated state (Fig. 2e) and eventually to the isolated hydrated electron (Fig. 2f). To identify the latter (blue region in Fig. 2h), we consider the flattening of

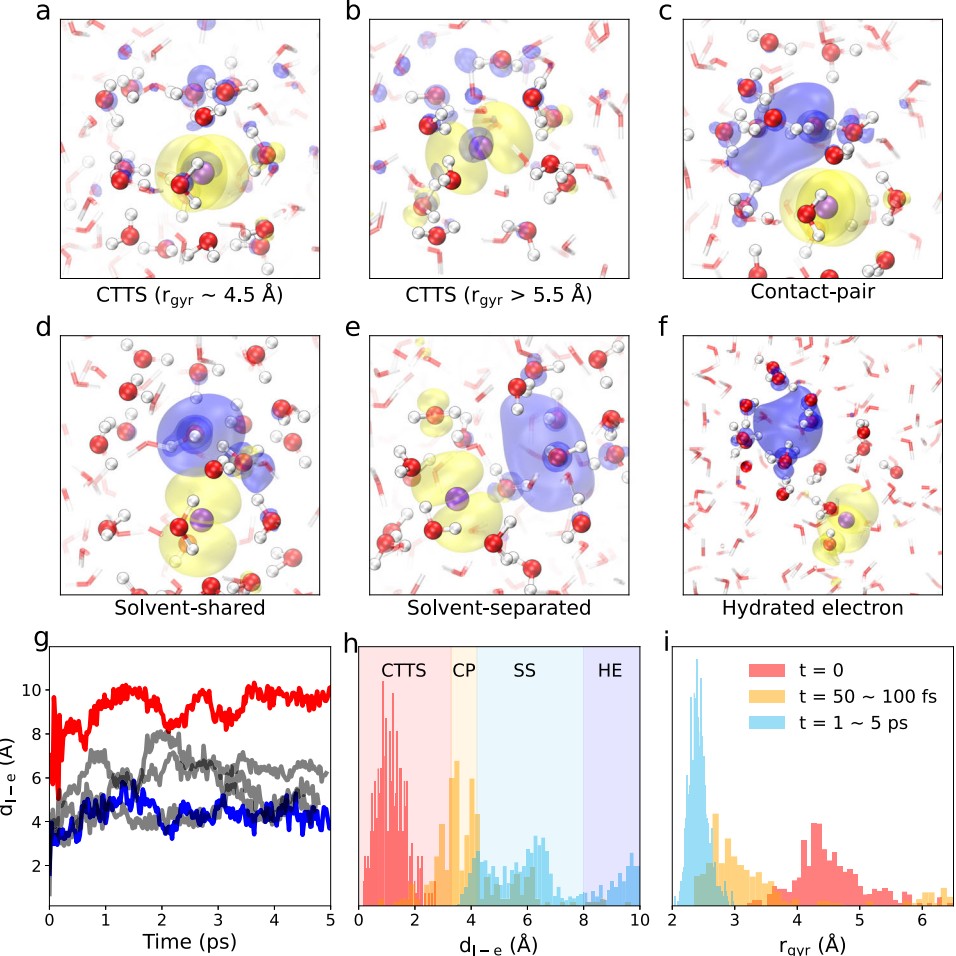

**Fig. 2 | CTTS dynamics from ab initio molecular dynamics. a, b** CTTS state; **c** contact-pair; **d** solvent-shared; **e** solvent-separated; **f** hydrated electron far from the aqueous iodine (color code: purple for iodine, red for oxygen, white for hydrogen, yellow for the hole density of aqueous iodine, and blue for the density of the excited electron). **g** I-$e_{aq}^-$ distance as a function of time for five trajectories, among which the cases where the hydrated electron is formed in the vicinity of the iodine (blue curve) and far from the iodine (red curve) are highlighted; (**h, i**) distribution of the I-$e_{aq}^-$ distances ($d_{I-e}$) and of the gyration radii ($r_{gyr}$) at various times upon electron excitation, where the structural configuration at $t = 0$ is obtained from the molecular dynamics trajectory of aqueous iodide. The shaded regions in **h** distinguish the CTTS (red), contact-pair (orange), solvent-separated (cyan), and hydrated electron (blue) states.

the free-energy profile at distances beyond 8.0 Å (vide infra, see Fig. 3a). Consequently, we take together the solvent-shared and solvent-separated states that we attribute to the interval between 4.3 and 8.0 Å (cyan region in Fig. 2h).

From our simulations, we see that most of the CTTS dynamics follow a reaction pathway in which the iodine-electron distance slowly increases with noticeable fluctuations in the range of 4–6 Å (blue and gray curves in Fig. 2g and Supplementary Movie 1). In one of these simulations (blue curve in Fig. 2g), we observe that the excited electron always remains in the vicinity of the iodine despite continuing the evolution for 25 ps (see Supplementary Fig. 8), likely due to the initial gyration radius being rather small. This is consistent with the occurrence of long-lived intermediates reaching a lifetime of up to 80 ps[5,21]. In another simulation (red curve in Fig. 2g), the aqueous iodine and the hydrated electron immediately form a separated state without going through contact-pair and solvent-separated states. Such a state arises from a delocalized CTTS state carrying a large gyration radius (Fig. 2a, e, f and Supplementary Movie 2). However, from our 50 CTTS simulations, we remark that only in a few cases such large distances are observed within 100 fs. These results are in line with the fluorescence up-conversion[18] and the TRPES[8,21,22] studies in that both report short-lived CTTS dynamics.

Our simulations also provide insight into the contentious mechanism of either trap-seeking or trap-digging[54,55]. In the trap-seeking mechanism, the electron searches for a pre-existing low-energy trap to occupy, while the trap-digging mechanism refers to an electron that creates a trap by forcing the rearrangement of the solvent. Our simulations indicate that the excited electron is initially formed in a state characterized by a wide range of gyration radii from 3.0 to 7 Å (see distribution at $t = 0$ in Fig. 2i) centered close to the iodine (see Fig. 1h, $t = 0$). For cases with gyration radii above 3.5 Å, which represent the vast majority, the electron density evolves generally to a contact-pair with a smaller gyration radius within 100 fs (see Fig. 2h and i). This process occurs through the reorientation of the OH groups and can be identified as trap-digging. In a small minority of cases, the hydrogen bonding in proximity of the iodine is loose. This gives initial gyration radii smaller than 3.5 Å and occurs in combination with rather large $d_{I-e}$ of 2 Å or more (cf. overlap between the red and orange regions in Fig. 2h and i). These rare cases correspond to a direct excitation to a contact-pair (Fig. 2c), indicating a trap-seeking mechanism. This observation could possibly explain the origin of the broad shoulder in the photokinetic energy[21] and the large vertical binding energies (VBEs) beyond 3 eV measured for the CTTS state (see red dashed-line in Fig. 3d)[22]. This indicates that the structure around

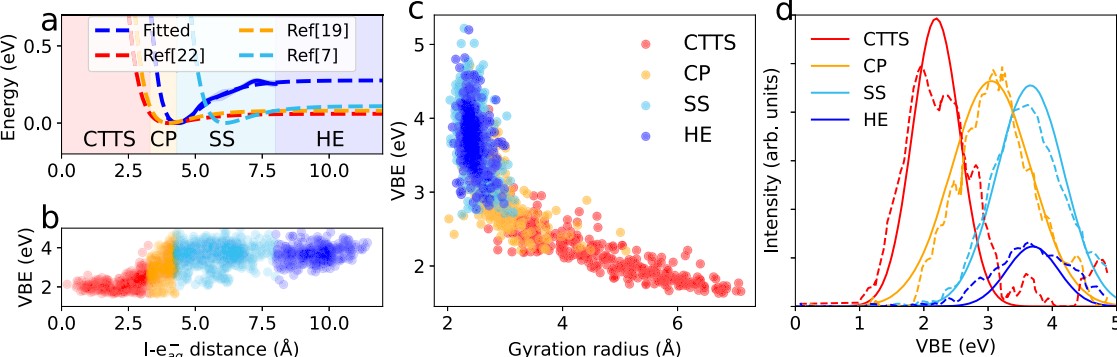

**Fig. 3 | Energetics of CTTS dynamics. a** Free-energy profile as a function of I-$e_{aq}^-$ distance, a Morse potential is fitted to the calculated free energy (blue dashed) to facilitate the comparison with experimental data from refs. 19 (orange dashed), [7] (cyan dashed), and [22] (red dashed). The shaded regions in **a** distinguish the CTTS (red), contact-pair (orange), solvent-separated (cyan), and hydrated electron (blue) states. **b** distribution of vertical binding energy (VBE) as a function of I-$e_{aq}^-$ distance. **c** distribution of VBE as a function of gyration radius of the excited electron. **d** calculated VBEs (solid) of charge-transfer-to-solvent (CTTS, red), contact-pair (CP, orange), solvent-separated (SS, cyan), and hydrated electron (HE, blue) states (solid), compared with experimental data taken from ref. 22 (dashed-line). The calculated VBEs of each state are fitted to a Gaussian function with the intensity normalized to the experimental data.

the iodide solute rarely forms pre-existing cavities that are able to attract the excited electron, in contrast with other solutions, such as methanol[56–58] and mixtures of n-tributyl phosphate and water[59], which typically exhibit trap-seeking behavior. However, we should remark that the gyration radii are continuously distributed from 3 to 6 Å, making a clear distinction between trap-seeking and trap-digging behaviors difficult. Notably, It is important to note that trap-seeking cases, where the electron localizes to a cavity structure near the iodine, still involve trap-digging in their dynamics, as one can see from the decreasing gyration radii as the electron further localizes (See Fig. 2h, i). Furthermore, we distinguish a few simulations, in which the initial electron densities show gyration radii of 6 Å or larger and give rise to the formation of hydrated electrons at large distances from the iodine. The cavity formation mechanism is then trap-digging in analogy with the case of neat water[54,60]. Hence, the trap-digging mechanism vastly dominates in our simulations suggesting the absence of pre-existing traps in most cases.

**Free energy**

To establish a connection with experimental results, we address the free energy of the excited electron during the CTTS process. The free-energy profile obtained from our simulations is displayed in Fig. 3a as a function of the I-$e_{aq}^-$ distance. A well-defined local minimum occurs at ~4.3 Å, which falls in the range of solvent-shared states (Fig. 2d). Our simulations indicate that the dissociation process of the iodine and the excited electron is endothermic by 0.25 eV, and does not require the overcoming of any significant barrier. At distances beyond 8 Å, the free-energy profile flattens out indicating that the hydrated electron state has been reached. It is convenient to fit the free-energy profile with a Morse potential:

$$V(r) = \Delta G[1 - e^{-\beta(r - r_{min})}]^2, \quad (1)$$

where $\Delta G$, $r_{min}$, and $\beta$ are parameters that describe the depth, the distance at the potential minimum, and the potential width, respectively. The parameters obtained from our simulation are given in Table 1, where they are compared with experimental counterparts extracted from TAS[7,19] and TRPES[22] measurements assuming a kinetic model based on the Smoluchowski diffusion equation (see Supplementary Note 7). To obtain the experimental parameters, it was necessary to fix the value of $r_{min}$ from the outset[7,19,22]. Iglev et al.[19] fixed this value at 6.1 Å by summing up the iodine-oxygen distance of 3 Å and the radius of 3.1 Å pertaining to the first solvation shell of the hydrated electron. At variance, Vilchiz et al.[19] and Yamamoto et al.[22] assumed that the

electrons are excited closer to the iodine and took a value of 4 Å for $r_{min}$. In our simulations, we find the local minimum of the potential at 4.3 Å lending support to the latter assumption. Furthermore, the potential width $\beta$ of 1.0 Å$^{-1}$ obtained from our simulations is consistently in excellent agreement with the experimental results obtained by setting $r_{min}$ to 4.0 Å (see Table 1)[19,22], but differs from the value of 0.7 Å$^{-1}$ obtained with $r_{min} = 6.1$ Å[7]. Indeed, from our simulations, we observe that the aqueous iodine, the excited electron, and the shared water molecule are not collinear in the solvent-shared state (see Fig. 2d). We find that the dissociation energy of 0.25 eV in our simulations overestimates typical values extracted from experimental data by ~0.1 eV. However, this discrepancy is sufficiently small to be attributed either to theoretical or experimental limitations. It is important to note that our simulation allows for the superposition of successive states of the CTTS process onto the free-energy profile (see Fig. 3a). Achieving a similar level of description with laser pump-probe experiments[9,19,21,22,61,62] is challenging because the experimental spectrum undergoes a continuous shift in energy without showing any major difference in shape as the underlying states evolve[22].

An important physical quantity characterizing the CTTS process as it evolves through its intermediate states is the vertical binding energy (VBE). To access the VBE of each molecular dynamics configuration, we monitor the Kohn-Sham energy level of the excited electron $E_{KS}$ with respect to that of the hydrated electron. We use the following relation:

$$VBE = E_{KS} - E_{KS}(e_{aq}^-) + VBE(e_{aq}^-) \quad (2)$$

where $VBE(e_{aq}^-)$ corresponds to the vertical binding energy determined for the hydrated electron in ref. 40 with the same computational set-up, while $E_{KS}(e_{aq}^-)$ is determined in our simulation by averaging over configurations in which the excited electron is found at large distances from the iodine (see blue dots in Fig. 3b). The reference value of the $VBE(e_{aq}^-)$ taken from ref. 40 has further been adjusted by 0.25 eV to account for nuclear quantum effects, as inferred from our present simulations (see Supplementary Fig. 11). This leads to a reference $VBE(e_{aq}^-)$ of 3.70 eV, in excellent agreement with the most recent experimental determination of 3.7 ± 0.1 eV[22,63]. In Fig. 3b, the calculated VBEs increase with the I-$e_{aq}^-$ distance, which we use to identify the intermediates of the CTTS process. At variance, the VBEs are found to decrease with gyration radius (Fig. 3c). Interestingly, despite the difference in I-$e_{aq}^-$ distance, the solvent-separated and hydrated electron states exhibit almost identical VBEs and gyration radii (cf. Fig. 3b and c). This observation suggests that the solvation shell

**Table 1 | Parameters of the Morse potential from our simulation and from transient absorption spectroscopy (TAS) and time-resolved photoelectron spectroscopy (TRPES) experiments**

| | TAS[19] | TAS[7] | TRPES[22] | Simulation |
|---|---|---|---|---|
| T (K) | 297 | 298 | 278 | 300 |
| $r_{min}$ ( Å) | 4.0 | 6.1 | 4.0 | 4.3 |
| $\beta$ ( Å$^{-1}$) | 1.1 | 0.7 | 1.0 | 1.0 |
| $\Delta G$ (eV) | 0.08 | 0.11 | 0.06 | 0.27 |

around the excited electron no longer depends on the iodine atom when they separate apart. The VBEs relate to the gyration radii through the $1/r_{gyr}^2$ dependence from the the particle-in-a-box model. This property has analogously been found to apply in the case of the hydrated electron in neat water[54,60,64]. To corroborate our state assignment, we compare in Fig. 3d the calculated VBEs of each intermediate with corresponding experimental data[22]. The calculated and measured VBEs for each state of the CTTS process are found to correspond closely. This agreement supports the validity of the state assignment put forward in our work. Furthermore, in the Supplementary Information, we also calculate time-resolved absorption spectra and extend the comparison to experimental fluorescence spectra[18].

In conclusion, our ab initio molecular dynamics simulations offer a comprehensive atomic-scale understanding of the CTTS dynamics. Upon electron excitation, the water molecules in the vicinity of the iodine atom readjust their orientation within a time scale on the order of a hundred femtoseconds. After this initial readjustment, the system undergoes further structural relaxation, which culminates in the formation of a contact-pair state, in which the iodine and the excited electron share the same solvation shell with I-$e_{aq}^-$ distances in the range of 3.3–4.3 Å. The distance of 4.3 Å marks a free-energy minimum, where the iodine and the excited electron share a water molecule without presenting individual solvation shells. Through the analysis of the structural properties and the free-energy profile, we have come to an assignment of the CTTS, contact-pair, solvent-separated, and hydrated electron states. In particular, our study provides a complete description of the contact-pair formation process at short times after electron excitation. The calculated vertical binding energies for these states are found in excellent agreement with experimental data, thereby supporting the validity of our assignment. Overall, our findings highlight the intricate interplay between solvent and solute during electron detachment and the essential role played by the solvent in promoting this fundamental process.

## Methods

### Hybrid-functional calculations
The underlying electronic structure is described with the spin-polarized hybrid functional PBEh (0.40)[37], in which the fraction of Fock exchange is set to 0.40. The functional is supplemented with van der Waals interactions through the nonlocal rVV10 functional[38], where the parameter $b$ is set to 5.3 to ensure that the structural and dynamical properties of liquid water are properly reproduced[39]. All the periodic AIMD calculations are performed with the CP2K code[65] employing triple-$\zeta$ basis sets[66] and Goedecker-Teter-Hutter pseudopotentials, which have been optimized against the PBE0 functional[67,68]. The exchange integrals are calculated with the auxiliary density matrix method[69].

### Molecular dynamics
To accelerate the simulations, we adopt the neural network scheme introduced by Behler and Parrinello[70], which is implemented in the n2p2 code[71]. The molecular dynamics simulations for aqueous iodine and aqueous iodide utilize machine learning potentials. We conduct

canonical ensemble (NVT) simulations at room temperature, employing a thermostat to ensure canonical sampling through velocity rescaling[72]. For the dynamics of quantum nuclei, we perform a thermostated ring polymer molecular dynamics with 32 beads[73]. The simulations are carried out with a timestep of 0.5 fs and last 50 ps each. In the CTTS simulation, we use a multi-timestep (MTS) scheme[44,45]. The machine learning potential acts as the fast component, while the hybrid functional serves as the slow component in the MTS-MD. Additionally, the densities of the excited electron and of the iodine radical are visualized through hybrid functional calculations. For the dynamics of the quantum nuclei, we combine the multiple-timestep scheme[44] and the ring polymer contraction method with 32 beads[45]. The machine learning potential is employed for both the fast component, utilized in the multiple-time step scheme, and the replica component, employed in the ring polymer contraction method. The slow component of the centroid, on the other hand, is governed by the hybrid functional. In the multiple-timestep scheme, the timestep for the fast component is set to 0.5 fs, while the slow component operates with a timestep of 2 fs.

## Data availability
The raw data for each figure and the Jupyter Notebook used to generate the figures are provided as source data files. Data generated and input files for this study that are not included in this article and its Supplementary Information are available at Materials Cloud: https://doi.org/10.24435/materialscloud:br-jf Source data are provided with this paper.

## Code availability
The computer codes used for electronic structure calculations (CP2K[65]), machine learning (n2p2[71]) and quantum dynamics (i-PI[74]) are described in the Supplementary Information.

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

## Acknowledgements

This work was supported by a grant from the Swiss National Supercomputing Centre (CSCS) under project ID s1123. J.L. thanks Simons Foundation Postdoctoral Fellowship.

## Author contributions

J.L., M.C., and A.P. conceived and designed research; J.L. performed research; J.L. wrote the first version of the manuscript. All authors contributed to the results analysis, discussion and manuscript revision.

## Competing interests

All authors declare no competing interests.
