## [Peer Review File · Nature Communications]

Dynamics of the Charge Transfer to Solvent Process in Aqueous IodideReviewer #1 (Remarks to the Author):

I have read the article entitled, "Charge Transfer to Solvent Dynamics in Aqueous Iodide" b" with great interest. It is a beautiful piece of simulation and theory and tells a detailed picture of the evolution of the CTTS to the solvated electron. It covers the extensive literature in a satisfactory fashion and provides insights into structure, electronic structure, free energetics and connects to experiment. These authors have been careful and the techniques and simulations are of high quality.

It largely corroborates the work of Carte-Fenk et al which is exciting given the difficulty of these calculations. It goes further in computing a glimpse into the free energetics which is is fascinating and important.

The article is well written and clear. It will make an excellent addition to Nature Communications and likely inspire many additional studies.

Reviewer #2 (Remarks to the Author):

In this paper, the authors present the most detailed simulation study to date of the CTTS process of aqueous iodide. The level of theory is currently state-of-the-art, and the results are tangentially consistent with known experimental findings. The fact that there are distinct contact, solvent-shared, solvent separated and completely separated iodine atom:hydrated electron pairs fits well with intuition developed from a whole host of previous ultrafast spectroscopy studies. I think this paper is a valuable contribution and recommend its publication in the literature subject to some of the comments presented below.

1) No error bars are presented on any of the figures. For example, Fig. 1 compares RDFs with and without quantum effects, but no error bars are provided, so it's not clear if the differences between the two sets of RDFs are meaningful. Why are the I-O RDFs more affected by the quantum treatment than the I-H RDFs? And without error bars, it's also hard to meaningfully compare the averages over the short trajectories vs. the equilibrium ensemble average.

2) In the section on dynamical properties section the authors examine I - e⁻ distances, but it's not clear what metric are they using to calculate the e⁻ position? Wannier function? Spin density COM? KS orbital COM? This should be specified.

3) It would be nice if the authors could specify what fraction of trajectories exhibit trap digging behavior vs. trap seeking.

4) On line 181, it would be nice if the authors could dig into this trajectory more and speculate as to why they didn't see the electron and iodine separate. Were any of the properties they investigated significantly different for this configuration compared to all of the others that did separate? They do speculate on the mechanism for the trajectory that skipped the contact-pair and solvent-separated states, so it would be helpful if they could comment on this 'anomalous' case as well.

5) On lines 250-251, the authors mention that "The reference value of the VBE(e⁻ - aq) taken from Ref. [40] has further been adjusted by 0.25 eV to account for nuclear quantum effects,..". Can they say more on where this adjustment comes from? It's not clear to me that ZPE from the H atoms would have this large an effect on the band gap given that the RDFs in Fig. 1 are so similar.

6) I was very skeptical of the idea of running dynamical trajectories in the triplet state until I saw the data in the SI that the energies and forces appear to be unaffected by the spin flip. Once the electron and I atom are separated, this makes perfect sense to me, but I have a hard time

understanding why this doesn't affect the energetics and dynamics in the CTTS Frank-Condon region. Can the authors explain this?

7) Although the information obtained from these simulations is outstanding, there is relatively little direct contact with experiment. Could the authors compute spectroscopy using TD-DFT for a small handful of their configurations in different places along the reaction coordinate and compare to the numerous transient absorption experiments that have been presented in the literature? In particular, the spectrum of the I atom-water complex should be quite unique and comparing to an experimental benchmark would be helpful. This would be a fair bit of work, but would make the paper much stronger.

Reviewer #3 (Remarks to the Author):

In this work, state of the art simulations are carried out to simulate a fundamental aqueous phase photo-induced reaction – charge separation on UV excitation of aqueous iodide, the so called CTTS process. The authors start by providing an excellent overview of the state of advancement of knowledge on CTTS dynamics and carefully position their work in the context of the rich and important literature of the iodide photoinduced charge separation. They apply ab initio molecular dynamics for the fully solvated system as well as machine learning approaches for the first time, to the knowledge of the reviewer. Despite the system's apparent simplicity, an atomic ion, these systems have defied the best theoretical tools because of the need to describe highly diffuse electronic states and the solvated electron photoproduct. As such this system as a benchmark is watched carefully by a broad community. This study is a significant and welcome advance and will be of considerable interest to a broad community of condensed phase spectroscopists, computation photochemists and theoreticians.

From the point of view of an experimentalist, the simulation results presented here are welcome, insightful and a significant addition to the literature. They heavily connect to past experimental attempts to define all aspects of this charge separation process, including the free energy surface for the distance of separation between the solvated electron and the iodine nucleus. There is a lot of excellent material here to propel future experimental work. While I defer to a theoretician reviewer as to the suitability of the methodologies employed, from my point of view the work is publishable as is and is suitable for Nature Comm.

Reply to Reviewers for the submitted manuscript

December 15, 2023

We would like to thank the Editor and the Reviewers for taking into consideration our work and suggesting improvements to the manuscript.

Detailed response to Reviewer 2

1) No error bars are presented on any of the figures. For example, Fig. 1 compares RDFs with and without quantum effects, but no error bars are provided, so it's not clear if the differences between the two sets of RDFs are meaningful. Why are the I-O RDFs more affected by the quantum treatment than the I-H RDFs? And without error bars, it's also hard to meaningfully compare the averages over the short trajectories vs. the equilibrium ensemble average.

We have indicated the error bars of RDFs in the revised Fig.1. The error bar had already been indicated in the Fig.3a.

Regarding the radial distribution functions (RDFs) in aqueous iodide, Fig. 1a and b illustrate that nuclear quantum effects (NQEs) have a more pronounced impact on I-H interactions compared to I-O. However, the disparity between quantum and classical dynamics is not markedly significant. Such a behavior might stem from a near-perfect compensation of opposing quantum effects, as often remarked in previous literature. For instance, such a phenomenon has also been observed in the context of liquid water [see Figure 5, Chem. Rev. 2016, 116, 7529–7550], water and heavy water [refer to Figure 2 for water and Table 1 for heavy water, Proceedings of the National Academy of Sciences 116.4 (2019): 1110-1115], and solvated electrons in heavy water [see Figure S4 in the Supporting Information, Nature Communications 12.1 (2021): 766].

Focusing on aqueous iodine, we observe that NQEs significantly influence the second solvation shell, particularly evident in the I-O interactions at approximately 4 Å. The reason NQEs distinctly affect I-O interactions at around 4 Å can be attributed to the hydrogen bond network in the I-H₂O complex. NQEs substantially modify the water structure within this complex, which in turn, alters the hydrogen bond network between the I-H₂O complex and surrounding water molecules. This leads to a significant impact on the RDFs around 4 Å.

In the revised version, from line 107 to 109, we mentioned the fact that the difference between the classical and quantum RDFs results from a compensation of opposing quantum effects. We also mentioned the NQEs contribution to I-H₂O complex from line 151 to 153.

2) In the section on dynamical properties section the authors examine I - e-distances, but it's not clear what metric are they using to calculate the e- position? Wannier function? Spin density COM? KS orbital COM? This should be specified.

For the excited electron, we obtain the density of the KS orbitals as ψ_{ks}^2 and calculate the corresponding gyration center and radius. The details have been provided in Section X of the Supporting Information. The same method for the treatment of the electron density has been used in our previous work. [Nature Communications 12.1 (2021): 766, Angew. Chem.Int. Ed.2022,61,e202209398]

In the revised manuscript, we add a short description from line 99 to 101. For the excited electron, we obtain the density of the Kohn-Sham orbital as ψ_{ks}^2 and calculate the corresponding gyration center and radius (See section X in SI)

3) It would be nice if the authors could specify what fraction of trajectories exhibit trap digging behavior vs. trap seeking.

We appreciate the interest in the distinction between trap digging and trap seeking behavior in our study. From our analysis of 50 independent charge transfer to solvent (CTTS) trajectories, we observed trends related to the initial gyration radius of the excited electron. Specifically, trajectories with a larger initial gyration radius tended to exhibit trap-digging behavior. In contrast, those with a smaller initial gyration radius were more inclined towards trap-seeking behavior. This pattern relates with the fluctuating structural configurations around the iodide at t=0, leading to either favorable or unfavorable conditions for electron incorporation in proximity of the iodide.

However, it is challenging to provide a precise fraction of trajectories showing each behavior, as these concepts are not well-defined. As Figure 3i (t=0, red) shows, the gyration radii are continuously distributed from 3 to 6 Å, without a clear separation to determine trap-seeking and trap-digging behaviors. It is important to note that trap-seeking cases, where the electron localizes to a cavity structure near iodine, still involve trap-digging in their dynamics. The smallest observed gyration radius, approximately 3 Å at t=0, is larger than that of a fully localized solvated electron (around 2.5 Å). Thus, for a fully hydrated structure, the digging process is essential. Additionally, there are instances where trap-seeking is absent from the dynamics, as addressed in point 4.

To sum up, the observed phenomenon can be explained as follows. When the structural configuration around the iodide allows it, the excited electron gets trapped into a localized state with a small gyration radius in proximity of the iodide, indicative of trap-seeking behavior. Conversely, when such immediate

favorable conditions are absent, the gyration radius of the excited electron is much larger and the electron needs to create a stable configuration to allow for its accommodation, thus displaying trap-digging behavior.

We further discuss the trap-seeking and trap-digging in the revised version from line 217 to 221.

4) On line 181, it would be nice if the authors could dig into this trajectory more and speculate as to why they didn't see the electron and iodine separate. Were any of the properties they investigated significantly different for this configuration compared to all of the others that did separate? They do speculate on the mechanism for the trajectory that skipped the contact-pair and solvent-separated states, so it would be helpful if they could comment on this 'anomalous' case as well.

In the majority of cases, the separation distance between the electron and iodine does not exceed 8 Å, even in our longest 25-ps trajectories (blue curve in Fig. 2 and Fig. S8). This observation is consistent with experimental findings, which suggest that the formation of a separated hydrated electron and iodine might take as long as 80 ps. Therefore, this particular trajectory referred to by reviewer doesn't stand out as an "anomalous" case. In the particular trajectory, the excited electron remain close to the iodine, likely because the initial gyration radius is relatively smaller.

In most simulations, the hydrogen bond network needs to adjust and reorient to form a cavity structure that can accommodate the excited electron, a process indicative of trap-digging behavior. The reorientations of the hydrogen bonds near the iodide competes with those in the bulk water region. In rare instances, the hydrogen bonds in the bulk water readjust more rapidly than those near the iodide. This leads to the mechanism that skips the CP and SS states.

In the revised manuscript, from line 188 to 191, we give the likely reason by which excited electron remains close to the iodine.

In one of these simulations (blue curve in Fig. 2g), we observe that the excited electron always remains in contact with the iodine despite continuing the evolution for 25 ps (see Fig. S8), likely due to the initial gyration radius being smaller.

5) On lines 250-251, the authors mention that "The reference value of the VBE(e - aq) taken from Ref. [40] has further been adjusted by 0.25 eV to account for nuclear quantum effects,.." can they say more on where this adjustment comes from? It's not clear to me that ZPE from the H atoms would have this large an effect on the band gap given that the RDFs in Fig. 1 are so similar.

We have compared the energy levels of e^- between our classical and quantum dynamics simulations. The Kohn-Sham orbital level in quantum dynamics is approximately 0.25 eV lower than that in classical dynamics. In the figure below, we present the Kohn-Sham orbital energy level of the e^- state and O_{2s} state as derived from both classical and quantum dynamics. The level has been aligned

with respect to the O_{2s} level, given the fact that the latter is deep level that can be taken to be constant with respect to the vacuum level. Consequently, the difference in the absolute energy level of e^- between classical and quantum dynamics is about 0.25 eV.

Kohn-Sham orbital energy level of the e^- state and O_{2s} state as derived from both classical and quantum dynamics

We agree that NQEs don't contribute a large effect on the RDFs of I-O or I-H. Notably, NQEs contribute significantly to the structure of water and of the hydrated electron, thereby affecting the energy level of electron. In our previous work, we have demonstrated that NQEs affect the structure of the hydrated electron. NQEs broaden the first peaks in both e^- -O and e^- -H, due to zero-point vibrational motion, and reduce considerably long-range order. [See Fig 1h,i in Nature Communications 12.1 (2021): 766]

In the revised manuscript, we have mentioned how this adjustment was obtained, as described in lines 264 to 266, and have provided additional details in Section VIII of the Supplementary Information.

6) I was very skeptical of the idea of running dynamical trajectories in the triplet state until I saw the data in the SI that the energies and forces appear to be unaffected by the spin flip. Once the electron and I atom are separated, this makes perfect sense to me, but I have a hard time understanding why this doesn't affect the energetics and dynamics in the CTTS Frank-Condon region. Can the authors explain this?

In the Supplementary Information (SI), we presented a benchmark comparison of the singlet state S1 from Time-Dependent Density Functional Theory (TDDFT) and the triplet state T1 from Unrestricted Kohn-Sham (UKS) methods, focusing on both energies and forces. Our benchmark results reveal an energy shift between the S1 and T1 states, while the forces remain almost identical. This similarity suggests that the shapes of the potential energy surfaces

(PES) for S1 and T1 are nearly indistinguishable. Since molecular dynamics trajectories are influenced more by the shape of the PES rather than their absolute energy levels, the dynamics derived from both T1 and S1 states are strikingly similar. This agreement stems from the large band gap (> 5 eV) between the two singly occupied molecular orbitals (SOMO-1 and SOMO), resulting in minimal mixing between the two states. This explains why the dynamics are not significantly affected in the CTTS Frank-Condon region.

We have included this part in the SI.

7) Although the information obtained from these simulations is outstanding, there is relatively little direct contact with experiment. Could the authors compute spectroscopy using TD-DFT for a small handful of their configurations in different places along the reaction coordinate and compare to the numerous transient absorption experiments that have been presented in the literature? In particular, the spectrum of the I atom-water complex should be quite unique and comparing to an experimental benchmark would be helpful. This would be a fair bit of work, but would make the paper much stronger.

We appreciate the suggestions to compare our results with transient absorption experiments. However, the signal observed in the transient absorption spectra do not directly reflect CTTS state, as the presence of solvated electrons heavily dominates these spectra. As a result, we have chosen to compare our theoretical spectra with experimental fluorescence spectra instead.

Specifically, we computed the absorption spectra at various stages along our CTTS trajectories. For these computations, we employed a model consisting of an iodine-water cluster with 70 water molecules surrounding the iodine atom. This quantum mechanical (QM) region was embedded within 560 point-charge water molecules, with the remaining environment modeled as a continuum solvation. All calculations were carried out using the ORCA code, employing the same functional and TZVP basis set.

The absorption spectra presented in the accompanying figure exhibit distinct characteristics at various time delays. In the left figure, we analyzed a single CTTS trajectory and calculated the absorption spectra from 0-100 fs using a time step of 2 fs. In the right figure, we averaged the spectra over three time intervals: 0-50 fs, 50-100 fs, and 500-1000 fs. Specifically, for the 500-1000 fs, we selected 50 snapshots evenly distributed across this time interval.

(Left) Time resolved absorption spectrum obtained with a TDDFT calculation. (Right) Absorption spectrum at different time windows, 0-50 fs, 50-100 fs and 500-1000 fs. (Bottom) Femtosecond fluorescence of aqueous iodine as taken from Fig. 1a of Ref.[in Nature Communications 4.1 (2013): 2119.]

It is important to note that our comparison is qualitative due to the Stokes shift between absorption and emission. The calculated spectra agree with previous experimental observations [refer to Fig.1a in Nature Communications 4.1 (2013): 2119]. Upon excitation of the iodide, the experimental fluorescence spectra presents a broad intensity distribution ranging from 280 to 650 nm, with a pronounced intensity peak near 340 nm. Correspondingly, our simulations in the time window 0-50 fs reveal a spectrum extending from 200 to 450 nm, with notable peaks at approximately 230 nm and 340 nm. In the time window 50-100 fs, the simulated spectrum undergoes a red shift, culminating in a significant peak near 400 nm. As the dynamics progress beyond 500 fs, both experimental and simulated spectra show a convergence towards a narrower intensity distribution and the formation of a single peak around 600 nm. These results offer valuable insights for future experiments, particularly highlighting the unique spectral features of the iodine-water complex formation as a function of time.

In the revised manuscript, we mentioned the TDDFT results in relation to experimental data towards the end, from lines 279 to 280, and provide a more detailed discussion in Section II of the Supplementary Information.

Reviewer #2 (Remarks to the Author):

Although there are still a lot of details that I think could have been better explained, the authors have given satisfactory responses to my previous comments, so I am happy to recommend publication in Nature Comm.